# The Loss of HJV Aggravates Muscle Atrophy by Promoting the Activation of the TβRII/Smad3 Pathway

**DOI:** 10.3390/ijms26052016

**Published:** 2025-02-26

**Authors:** Lu Wang, Wuchen Tao, Jiajie Jia, Min Yuan, Wenjiong Li, Peng Zhang, Xiaoping Chen

**Affiliations:** 1National Key Laboratory of Human Factors Engineering, China Astronaut Research and Training Center, Beijing 100094, China; winterangle@163.com (L.W.); 18363424228@163.com (W.T.); jiajiajie@bsu.edu.cn (J.J.); 2Department of Exercise Physiology, Beijing Sport University, Beijing 100084, China; 3National Key Laboratory of Space Medicine, China Astronaut Research and Training Center, Beijing 100094, China; y_min5628@163.com (M.Y.); muzijiong2007@163.com (W.L.)

**Keywords:** hemojuvelin, muscle atrophy, hindlimb unloading, TβRII, Smad3, ubiquitin ligases

## Abstract

Hemojuvelin (HJV) is a membrane-bound protein prominently expressed in the skeletal muscle, heart, and liver. Despite its established function in iron regulation, the specific role of HJV in muscle physiology and pathophysiology is not well understood. In this study, we explored the involvement of HJV in disuse-induced muscle atrophy and uncovered the potential mechanisms. Hindlimb unloading (HU) resulted in soleus muscle atrophy in wild type (WT) mice, accompanied by a significant decrease in HJV protein expression. The muscle-specific deletion of *Hjv* (MKO) exacerbated myofiber atrophy, which was associated with an increase in the expression of muscle ubiquitin ligases following HU. Furthermore, the expression of transforming growth factor-β type II receptor (TβRII) and the level of phosphorylated Smad3 (p-Smad3) were elevated after HU, and these effects were exacerbated in MKO mice. The knockdown of TβRII in the skeletal muscle of MKO mice mitigated myofiber atrophy and reversed the hyperactivation of the TβRII/Smad3 pathway induced by HU. Our findings demonstrate that the absence of HJV contributes to the activation of the TβRII/Smad3 signaling pathway and, consequently, the onset of myofiber atrophy in response to HU. Given its abundant expression in skeletal muscle, HJV emerges as a potential therapeutic target for muscle atrophy.

## 1. Introduction

The maintenance of muscle mass depends on the equilibrium between protein synthesis and degradation. When protein breakdown exceeds synthesis under dysregulated catabolic conditions (e.g., muscle disuse, sepsis, cancer cachexia, diabetes, congestive heart failure, and aging), muscle atrophy occurs [1]. The activation of the ubiquitin–proteasome system (UPS) accounts for most of the contractile protein degradation observed in muscle atrophy [2,3]. E3 ubiquitin ligases in the UPS play a vital role in controlling the proteolytic rate by catalyzing ubiquitination, a process that involves the attachment of ubiquitin molecules to specific lysine residues in substrate proteins [2,3]. The expression of two muscle-specific E3 ubiquitin ligases—muscle ring finger1 (MuRF1) and Atrogin-1 (also known as muscle atrophy F-box [MAFbx])—were reported to be significantly upregulated in multiple models of muscle atrophy [4,5,6]. Both Atrogin-1 and MuRF1 can target muscle contractile proteins for degradation, including myosin light chain, myosin heavy chain, and troponin I [6], thereby contributing to the loss of muscle mass and a decrease in muscle force.

The transforming growth factor-beta (TGF-β)/Smad3 signaling pathway has been implicated in a variety of muscle-wasting conditions [7,8,9,10,11]. The overexpression of TGF-β1 or myostatin, both members of the TGF-β superfamily, along with their type I receptors, such as activin receptor-like kinase 5, can induce muscle atrophy through the activation of the downstream mediators Smad2/3 [12,13]. The in vivo overexpression of Smad3 has been shown to upregulate Atrogin-1 expression, triggering myofiber atrophy by suppressing the protein kinase B (AKT)/mammalian target of rapamycin (mTOR) signaling pathway and protein synthesis [13,14,15]. The levels of Smad2/3 phosphorylation are increased in disuse-induced muscle atrophy, while the loss of Smad2/3 in muscles confers significant resistance to atrophy and [16] inhibits atrogene upregulation [16,17]. Interestingly, the absence of myostatin in adult mice does not prevent the activation of Smad2/3 or muscle atrophy due to disuse, suggesting that the atrophic process is more likely to be regulated at the receptor level rather than at the ligand level [16].

Hemojuvenin (HJV), also known as repulsive guidance molecule family member c (RGMc), belongs to the RGM family, the members of which serve as coreceptors for TGF-β signaling [18,19,20]. While HJV is highly expressed in skeletal muscle, it is best known for its role in systemic iron homeostasis in the liver as a coreceptor for bone morphogenetic proteins (BMP) [20,21]. Our recent research uncovered a novel function for HJV as a coreceptor for TGF-β type II receptor (TβRII) in skeletal muscle, influencing TGF-β1/Smad3 signaling in both physiological and pathological contexts [22]. Mice with systemic or muscle-specific *Hjv* knockout (MKO) exhibit muscle atrophy along with elevated expression of the E3 ubiquitin ligases MuRF1 and Atrogin-1. Furthermore, HJV expression is significantly reduced in the muscles of patients with Duchenne muscular dystrophy (DMD), *mdx* (DMD model) mice, and aging humans and mice. The overexpression of HJV mitigates dystrophy- and aging-related muscle wasting by inhibiting TGF-β-Smad2/3 signaling [22]. However, whether HJV is involved in disuse-induced muscle atrophy and Smad2/3 activation remains unknown.

Given the critical role of the TGF-β/Smad3 signaling pathway in muscle atrophy and the emerging evidence of HJV’s function as a coreceptor for TβRII, we hypothesized that HJV may serve as a key regulator of disuse-induced muscle atrophy by modulating the TβRII/Smad3 pathway. To test this hypothesis, we established a hindlimb unloading (HU) model to induce disuse muscle atrophy in both MKO mice and their wild-type (WT) littermates. By comparing the extent of muscle atrophy and associated molecular alterations between these genotypes, we sought to determine the role of HJV in disuse-induced muscle atrophy and uncover the potential receptor-mediated mechanism through which HJV may influence this process.

## 2. Results

### 2.1. HJV Expression Was Decreased in Atrophied Muscles

To explore the role of HJV in disuse-induced muscle atrophy, we monitored HJV levels in atrophied muscles following HU. After 14 days of HU, we observed substantial atrophy in the hindlimb muscles of WT mice, with significant loss of mass in both the soleus and gastrocnemius muscles (Figure 1A). Additionally, the expression of both *Atrogin-1* and *MuRF1* was increased in atrophied muscles after HU (Figure 1B), accompanied by a notable decrease in HJV protein levels (Figure 1C). These data suggested that a reduction in HJV protein levels may contribute to disuse-induced muscle atrophy.

### 2.2. The Depletion of Hjv Exacerbated Disuse-Induced Muscle Atrophy

To confirm the involvement of HJV in disuse-induced muscle atrophy, MKO mice were subjected to HU, with their *Hjv^flox/flox^* littermates serving as WT controls. Following 14 days of HU, WT mice exhibited a marked decrease in muscle mass and myofiber cross-sectional area (CSA), along with an increase in Atrogin-1 and MuRF1 expression (Figure 2). Compared with WT controls, the absence of *Hjv* resulted in only a minor reduction in the muscle mass of untreated mice, which was not statistically significant. However, a notable difference in both soleus and gastrocnemius muscle mass was detected between WT and MKO mice following HU, suggesting that MKO mice experienced greater muscle mass loss than their WT littermates (Figure 2A). The muscle fiber CSA in untreated MKO mice was comparable to that of unloaded WT mice, indicative of myofiber atrophy, as previously reported by us [22]. Moreover, the absence of *Hjv* led to a further decrease in CSA after HU (Figure 2B). The expression of Atrogin-1 and MuRF1 in MKO muscles was higher than that in WT muscles before HU, reaching levels similar to those seen in unloaded WT mice, and was further increased after HU (Figure 2C). Together, these results suggest that the muscle-specific knockout of *Hjv* aggravates HU-induced muscle atrophy by upregulating the expression of Atrogin-1 and MuRF1.

### 2.3. The Depletion of Hjv Enhanced Muscle Disuse-Induced Smad3 Activation by Upregulating TβRII Expression

We previously established that HJV inhibits TGF-β signaling by functioning as an inhibitory coreceptor for TβRII [22]. Thus, we next sought to ascertain whether the depletion of *Hjv* could enhance Smad3 activation in muscles subjected to HU. As depicted in Figure 3, HU led to elevated levels of both TβRII and p-Smad3 in WT mice. *Hjv* deficiency resulted in a further increase in TβRII and p-Smad3 levels in the unloaded muscles, indicative of TβRII/Smad3 pathway overactivation. Furthermore, when TβRII expression was suppressed in WT muscles using short hairpin RNA (shRNA), the HU-induced activation of p-Smad3 was greatly inhibited (Figure 4). Additionally, the knockdown of *TβRII* reversed the overactivation of p-Smad3 in the muscles of MKO mice to levels comparable to those of WT mice subjected to HU and treated with control Adv (Figure 4). This suggested that the reduction in HJV levels caused by HU contributes to the activation (phosphorylation) of Smad3 in muscle by upregulating TβRII expression.

### 2.4. The Knockdown of TβRII Reversed the Pro-Atrophic Effects Observed in MKO Mice

To determine if TβRII mediates the pro-atrophic effects seen in MKO mice, we investigated the impact of *TβRII* knockdown on muscle atrophy. As depicted in Figure 5, the knockdown of *TβRII* in WT mice mitigated the muscle mass loss and the increase in the expression of Atrogin-1 and MuRF1 induced by HU. Additionally, *TβRII* knockdown in MKO mice subjected to HU resulted in a muscle mass reduction similar to that observed in unloaded WT mice injected with control shRNA (Figure 5A). Furthermore, the elevated levels of Atrogin-1 and MuRF1 in HU-treated MKO mice were significantly reduced by *TβRII* knockdown, aligning with the levels observed in unloaded WT mice treated with control shRNA (Figure 5B). In conclusion, these findings indicate that the absence of *Hjv* exacerbates disuse-induced muscle atrophy primarily by upregulating the expression of TβRII.

## 3. Discussion

HJV, a member of the RGM family, is well-known for its role in systemic iron homeostasis through the modulation of hepcidin expression as a coreceptor for BMP in the liver [18,19,20]. The expression of HJV exhibits some tissue selectivity, being found mainly in skeletal muscle, the liver, and the heart, with the highest expression detected in skeletal muscle [21]. As the promoter region of the *Hjv* gene contains two evolutionary conserved E-boxes and a MEF2 site, *Hjv* mRNA is highly induced and parallels the upregulation of myogenin during myogenic differentiation [23,24]. This indicates that HJV may play a vital role in both physiological and pathophysiological processes in muscle. Indeed, we previously identified a new function of HJV involving the maintenance of muscle mass and function [22]. We found that both systemic and muscle-specific *Hjv*-knockout mice displayed myofiber atrophy due to the upregulation of MuRF1 and Atrogin-1 [22]. In addition, HJV expression was reported to be decreased in dystrophic and aged muscles, while its overexpression mitigated dystrophy- and age-related muscle wasting [22]. In this study, we also found that HJV expression was reduced in atrophied muscles due to disuse, and muscle-specific *Hjv* knockout aggravated disuse-induced muscle atrophy by promoting the expression of Atrogin-1 and MuRF1. Thus, the loss of HJV in skeletal muscle may represent an initiating event in the cascade of atrophic responses to various physiological or pathological stimuli.

Members of the TGF-β superfamily, including TGF-β1, myostatin, and activins A and B, are potent inhibitors of muscle mass and have been implicated in muscle atrophy in numerous relevant models [7,8,9,10,11,12]. For example, TGF-β1 expression is increased in the muscles of older adults and those of patients with cachexia, DMD, and amyotrophic lateral sclerosis, and contributes to muscle atrophy under these conditions [7,8,9,10]. Both systemic and local administration of TGF-β1 can induce muscle atrophy by upregulating the expression of the muscle-specific E3 ubiquitin ligases Atrogin-1 and MuRF1 [11,12,25]. However, disuse-induced muscle atrophy seems to be independent of systemic levels of TGF-β1 or other TGF-β superfamily ligands, as this type of atrophy only occurs in affected muscles. It has been reported that the absence of myostatin in adult muscles does not prevent disuse-induced muscle atrophy or alleviate Smad2/3 activation [16,17]. These observations suggest that disuse-induced muscle atrophy and Smad2/3 activation might be regulated at the receptor level rather than at the ligand level.

In this study, we found that the loss of HJV due to HU is an important membrane-receptor-level mechanism that can trigger downstream Smad3 activation, leading to muscle atrophy. We found that HJV deficiency resulting from HU was closely associated with elevated expression of TβRII and the activation of Smad3 in atrophied muscles. Furthermore, the knockout of *Hjv* in skeletal muscle enhanced the HU-induced upregulation of TβRII and activation of Smad3. We previously demonstrated that HJV is an inhibitory coreceptor for TβRII [22]. In vitro experiments confirmed that the loss of *Hjv* not only increased the basal levels of p-Smad3 and CAGA luciferase activity in C2C12 cells but also promoted the activation of Smad3 signaling induced by TGF-β1. In contrast, *Hjv* overexpression was sufficient to inhibit TGF-β1/Smad3 signaling both in vivo and in vitro [22]. Based on the current evidence, we propose two potential mechanisms by which HJV may modulate TGF-β signaling, namely either through direct interference with the molecular interaction between TGF-β ligands and TβRII, or via competitive binding to TβRII that subsequently disrupts type I receptor recruitment and activation [26]. In the present study, silencing the *TβRII* gene reversed the pro-atrophic effect observed in HU-treated *Hjv*-knockout mice, indicating that the loss of HJV due to HU may heighten muscle cell sensitivity to TGF-β1 by upregulating TβRII expression, leading to hyperactivation of downstream Smad3 signaling and, ultimately, the exacerbation of muscle atrophy. Interestingly, receptor-level regulation is also supported by evidence from studies on the IGF-1 pathway in disused muscles. The overexpression of IGF-1 in skeletal muscle does not prevent disuse-induced atrophy as the IGF-1 receptors in disused muscles are insensitive to circulating IGF-1 and are not easily activated [16,27,28]. In addition, the absence of IGF-1 receptors in muscles can significantly reduce muscle volume [28]. The inactivation of these receptors due to disuse leads to the inhibition of AKT phosphorylation and a decrease in protein synthesis, thereby contributing to muscle atrophy.

Smad3 is a key regulator of the transcriptional program that induces muscle atrophy [14,15,16,17]. Smad3 activation has been reported in muscle atrophy induced by a variety of conditions, including unloading [29], immobilization [17], denervation [23], sepsis [24], dystrophy, and aging [22,30]. The muscle-specific knockout of Smad3 was shown to attenuate disuse-induced skeletal muscle atrophy and inhibit the HU-induced upregulation of atrogene expression [16,17]. Mechanistically, Smad3 activation can inhibit the AKT/mTOR pathway, leading to reduced protein synthesis in skeletal muscle [14]. Meanwhile, Smad3 activation can increase FoxO3 expression, which directly regulates the levels of the ubiquitin ligases MuRF1 and Atrogin-1 [14,17]. In vitro experiments demonstrated that the overexpression of Smad3 alone is not sufficient to upregulate the expression of MuRF1 and Atrogin-1; however, Smad3 overexpression can synergistically augment FoxO3-induced Atrogin-1 and MuRF1 expression by increasing FoxO3 abundance at the promoter regions of these genes [14,15]. Although we did not assess the expression of AKT/FoxO3 in this study, it is likely that the hyperactivation of Smad3 induced by *Hjv* knockout led to the inactivation of AKT and FoxO3, thereby enhancing muscle-specific ubiquitin ligase expression in HU-treated muscles.

One limitation of our study is that we did not investigate whether iron dysregulation contributes to the enhanced effect of *Hjv* knockout on disuse-induced muscle atrophy. Iron dysregulation is increasingly recognized as a contributing factor to various types of muscle atrophy, acting through mechanisms involving oxidative stress, autophagy, and iron regulatory proteins [31,32,33,34,35]. Therapeutic strategies targeting iron metabolism, such as iron chelators or iron supplementation, have shown promise in mitigating muscle atrophy associated with aging or cachexia [36,37]. During the development of disuse-induced muscle atrophy, an increase in iron concentration has been observed, particularly in female mice [38]. Although skeletal muscle-specific knockout of *Hjv* does not affect systemic or muscle iron concentrations under normal ground-based conditions [39], it remains unclear whether *Hjv* deficiency exacerbates iron dysregulation during disuse conditions, potentially amplifying muscle atrophy. This represents a critical gap in our understanding that warrants further investigation to fully elucidate the interplay between *Hjv* knockout, iron homeostasis, and disuse-induced muscle atrophy progression. Another limitation in this study is the lack of a power analysis during the experimental design phase. Power analysis is an important tool for ensuring the robustness of statistical conclusions. Despite the lack of a formal power analysis, we believe that our study provides valuable insights due to the high-quality data collected and the comprehensive analysis performed.

## 4. Materials and Methods

### 4.1. Mice and Hindlimb Unloading

Exons 2 to 4 of the murine *Hjv* gene were floxed between two loxP sites using CRISPR/Cas9-mediated genome engineering, generating a novel *Hjv^flox/flox^* mouse line (Appendix A) (GemPharmatech Co., Ltd.; Nanjing, Jiangsu, China). *Hjv^flox/flox^* mice were then crossed with mice of the B6.Cg-Tg(ACTA1-cre)79Jme/J transgenic strain (cre recombinase gene driven by human alpha-skeletal actin [HSA or ACTA1]; The Jackson Laboratory, Bar Harbor, ME, USA), generating a muscle-specific *Hjv*-knockout line (*HSA-Cre; Hjv^flox/flox^* mice, MKO), as previously described (Appendix A) [22]. The primers used for *genotyping* are described in Appendix A.

All mice were maintained on a standard chow diet at a constant temperature (20 °C) under a 12 h/12 h artificial light/dark cycle. At 3 months of age, male mice were either subjected to HU or left untreated (controls) as previously described [30]. Briefly, the hindlimbs of the mice were completely suspended off the cage floor with a body inclination of approximately 30°, while the forelimbs remained in contact with a grid floor, allowing the animals a full range of motion [30]. Food and water were available ad libitum. Control mice were maintained free without HU in single cages. In some experiments, the hindlimb muscles were injected with adenovirus (Adv; described below) 3 days before HU. After two weeks of HU, the hindlimb muscles were removed, weighed, and either frozen in liquid nitrogen for subsequent protein extraction or embedded in a tissue-freezing medium for immunohistochemical analysis.

### 4.2. RNA Extraction and Real-Time PCR (qPCR)

Total RNA was isolated from each muscle sample using TRIzol reagent (Thermo Scientific, Waltham, MA, USA) and 2 μg of total RNA was reverse-transcribed into complementary DNA using a PrimeScript RT reagent kit (TaKaRa, Kusatsa, Shiga, Japan) according to the manufacturer’s guidelines. qPCR was conducted in triplicate using the StepOnePlus Real-time PCR system (Thermo Scientific). The PCR mixture, with a final volume of 50 μL, consisted of 21 μL of sterile water, 25 μL of SYBR Green (Thermo Scientific), 2 μL of cDNA, and 1 μL each of forward primer and reverse primer (10 pmol/μL). The PCR cycling conditions were an initial denaturation at 95 °C for 10 min, followed by 40 cycles of 95 °C for 10 s, 62 °C for 15 s, and 72 °C for 20 s. Melting curve analysis was conducted at 70 °C for 15 s, with continuous fluorescence monitoring as the temperature was increased from 60 to 95 °C at intervals of 0.3 s. Relative gene expression levels were determined using the comparative CT (2^−ΔΔCt^) method and normalized to those of the control group. The experiments were independently performed at least three times. The primer sequences are detailed in Appendix A.

### 4.3. Adenoviral Vector Construction and In Vivo Injection

The recombinant adenoviral shuttle vector pAdtrack-CMV-GFP carrying *TβRII* shRNA (Ad-shTβRII) was constructed by GeneChem Co., Ltd. (Shanghai, China). Adenovirus-mediated silencing was performed as previously described. The sequences of the shRNAs for *TβRII* were designed based on the GenBank accession number NM_009371.3. The optimal shRNA sequence selected for knockdown was 5′-caTGGAAGAGTGCAACGATTA-3′. No significant homology to known mouse gene sequences was found for this sequence in the GenBank database. The scrambled sequence used as a negative control was 5′-TTCTCCGAACGTGTCACGT-3′. Ad-shTβRII was injected in vivo (2 × 10^10^ PFU/mL) into the gastrocnemius muscle of MKO and WT mice, while the contralateral gastrocnemius was injected with Ad-shScramble (2 × 10^10^ PFU/mL) in a final volume of 20 μL.

### 4.4. Western Blot Analysis

Skeletal muscles were homogenized in lysis buffer (25 mM Tris·HCl [pH 7.6], 150 mM NaCl, 1 mM EDTA, 1% Triton X-100, and 10% glycerol) supplemented with a 1% protease inhibitor cocktail (Roche, Mannheim, Germany). The supernatants were collected, and the protein concentrations were determined using the BCA method (Thermo Scientific). Equal amounts (30 µg) of extracted protein were denatured in SDS loading buffer and separated on SDS–polyacrylamide gels. The protein was then transferred onto a nitrocellulose membrane, blocked in 5% nonfat milk diluted in TBS–Tween for 2 h, and incubated overnight at 4 °C with primary antibodies against HJV (R&D Systems, Minneapolis, MN, USA), TβRII (Abcam, Cambridge, MA, USA), Smad3 (Cell Signaling Technology, Danvers, MA, USA), p-Smad3 (Cell Signaling Technology), Atrogin-1 (ECM Biosciences, Versailles, KY, USA), MuRF1 (ECM Biosciences), and β-actin (Santa Cruz Biotechnologies, Dallas, TX, USA). Signals were visualized by incubating the samples with HRP-coupled secondary antibodies and enhanced chemiluminescence reagent (Thermo Scientific). If necessary, blots were stripped using stripping reagent (Thermo Scientific) according to the manufacturer’s instructions and then re-probed. Protein signals were quantified using Image-Pro Plus 6.0 and normalized to the corresponding β-actin signal. Data are expressed as a percentage of the expression levels in the control group.

### 4.5. Immunohistochemical Analysis

Cryosections of hindlimb muscles were incubated for 30 min in 0.3% Triton X-100 in PBS followed by blocking for 1 h in 5% goat serum in PBS. The CSA of hindlimb muscle fibers was determined via immunostaining with an anti-laminin antibody (Abcam). Alexa Fluor 594-conjugated goat anti-rabbit IgG (H + L) (Thermo Scientific) served as the secondary antibody. The CSA of fibers was calculated using Image-Pro Plus 6.0.

### 4.6. Statistical Analysis

Data are presented as means ± standard deviation (SD). Student’s *t*-test was used for comparisons between two groups. Analysis of variance (ANOVA) with a post hoc Tukey’s test was applied for comparisons among three or more groups as indicated in the figure legends. *p*-values below 0.05 were considered statistically significant.

## 5. Conclusions

In summary, in this study, we uncovered a novel receptor-mediated mechanism underlying disuse-induced muscle atrophy. Specifically, we found that muscle disuse leads to a decrease in HJV expression in skeletal muscle, which, in turn, activates the TβRII/Smad3 pathway. Smad3 activation promotes the expression of muscle-specific E3, resulting in muscle wasting. The inhibition of the TGF-β1/Smad3 signaling pathway represents a promising strategy for ameliorating muscle wasting disorders; however, how to manipulate this pathway in a tissue-specific manner remains to be determined. The high expression of HJV in skeletal muscle and its role in mediating disuse-induced muscle atrophy via the regulation of TGF-β1/Smad3 signaling make it a promising therapeutic target for muscle wasting disorders.

## Figures and Tables

**Figure 1 ijms-26-02016-f001:**
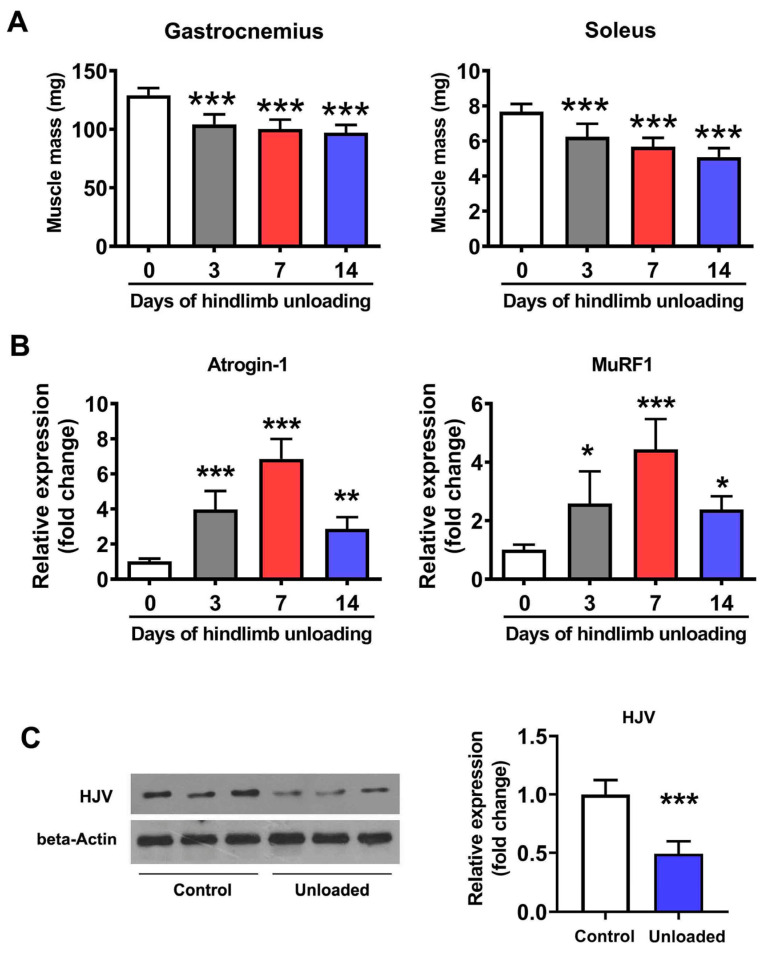
HJV expression is decreased in the soleus muscle of mice following HU. (**A**) Changes in skeletal muscle mass during 14 days of HU (*n* = 6). (**B**) qPCR analysis of *Atrogin-1* and *MuRF1* expression levels in the soleus muscle of mice during 14 days of HU (*n* = 6). (**C**) Western blot analysis of HJV protein expression in the soleus muscle after HU (*n* = 6). Data are shown as means ± SD. * *p* < 0.05, ** *p* < 0.01, *** *p* < 0.001 vs. the control (0) group by one-way ANOVA followed by a post hoc Tukey’s test (**A**,**B**) or *t*-test (**C**).

**Figure 2 ijms-26-02016-f002:**
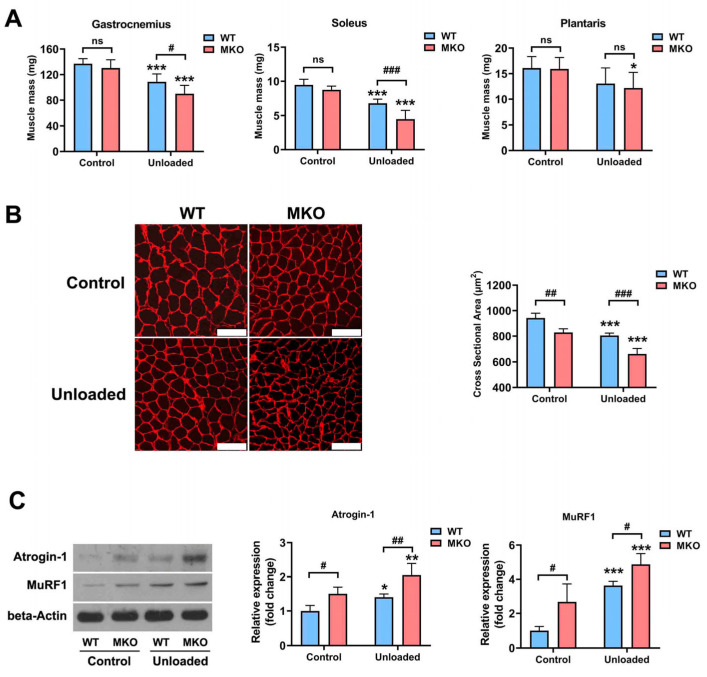
*Hjv* knockout in skeletal muscle promotes HU-induced muscle atrophy. (**A**) Skeletal muscle mass in control and unloaded WT and MKO mice (*n* = 8). (**B**) Representative image of immunostaining for laminin in soleus muscle cross-sections from control and unloaded WT and MKO mice and quantification of the CSA (*n* = 4). Scale bars: 100 μm. (**C**) Western blot analysis of MuRF1 and Atrogin-1 expression in the soleus muscle of control and unloaded WT and MKO mice (*n* = 5). Data are shown as means ± SD. * *p* < 0.05, ** *p* < 0.01, *** *p* < 0.001 vs. the control group of the same genotype; # *p* < 0.05, ## *p* < 0.01, ### *p* < 0.001 vs. as indicated (ANOVA followed by a post hoc Tukey’s test). MKO, muscle-specific *Hjv*-knockout mice; ns, no significance; WT, wild type.

**Figure 3 ijms-26-02016-f003:**
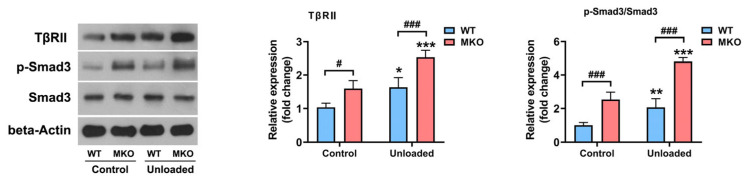
*Hjv* knockout promotes HU-induced Smad3 activation. Western blot analysis of Smad3, p-Smad3, and TβRII levels in the skeletal muscles of control and unloaded WT and MKO mice (*n* = 4). Data are shown as means ± SD. * *p* < 0.05, ** *p* < 0.01, *** *p* < 0.001 vs. the control group of the same genotype; # *p* < 0.05, ### *p* < 0.001 vs. as indicated (ANOVA followed by a post hoc Tukey’s test). MKO, muscle-specific *Hjv*-knockout mice; WT, wild type.

**Figure 4 ijms-26-02016-f004:**
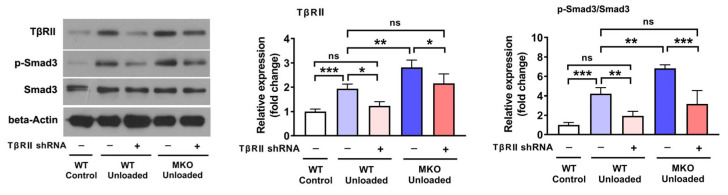
*TβRII* knockdown reverses the HU-induced Smad3 overactivation in MKO mice. Western blot analysis of Smad3, p-Smad3, and TβRII levels in skeletal muscles injected with the indicated construct (*n* = 4). Data are shown as means ± SD. * *p* < 0.05, ** *p* < 0.01, *** *p* < 0.001 vs. as indicated (ANOVA followed by a post hoc Tukey’s test). MKO, muscle-specific *Hjv*-knockout mice; ns, no significance; WT, wild type.

**Figure 5 ijms-26-02016-f005:**
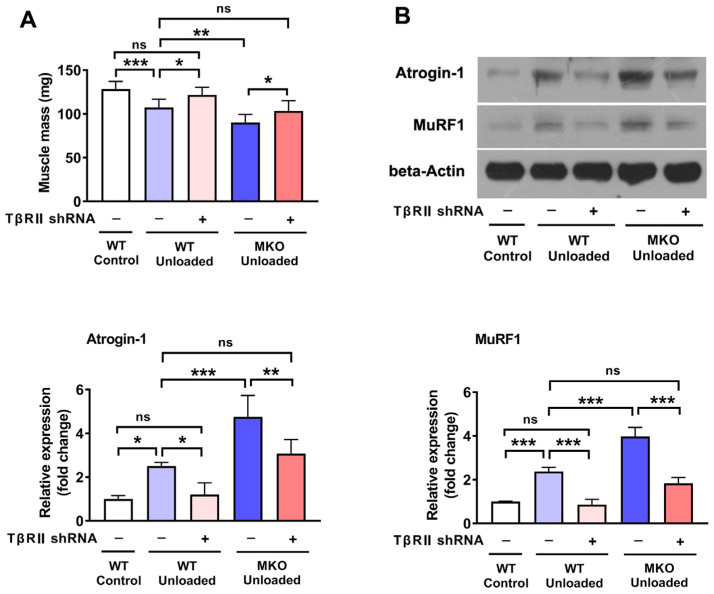
The knockdown of *TβRII* reverses the pro-atrophic effects observed in MKO mice. (**A**) Muscle mass of gastrocnemius muscles injected with the indicated constructs (*n* = 9). (**B**) Western blot analysis of MuRF1 and Atrogin-1 in muscle injected with the indicated constructs (*n* = 4). Data are shown as means ± SD. * *p* < 0.05, ** *p* < 0.01, *** *p* < 0.001 vs. as indicated (ANOVA followed by a post hoc Tukey’s test). MKO, muscle-specific *Hjv*-knockout mice; ns, no significance; WT, wild type.

## Data Availability

The data on which this article are based are available in the article and in its online Appendix A. Further inquiries can be directed to the corresponding author.

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
