# Peer review of "The Loss of HJV Aggravates Muscle Atrophy by Promoting the Activation of the TβRII/Smad3 Pathway"

_ijms, 2025, doi:10.3390/ijms26052016_

Round 1

Reviewer 1 Report

Comments and Suggestions for Authors

I congratulate the authors for the article. It is both relevant and of scientific significance.

Please find below my specific comments. 

Introduction: it is well written and elaborated until the last paragraph. The ultimate introduction paragraph seems to be accidentally copy-pasted from the conclusion part!!! Therefore this paragraph needs to be deleted. Instead, another paragraph should be added. This new (final introduction) paragraph should provide information regarding the study aims and hypothesis.

Methods: this section is well-written, comprehensive, and reproductious. Nonetheless, I think a power analysis is needed to be conducted in order to describe if the study sample. 

Additionally, ANOVA is mentioned as the statistical test used to compare groups. Yet, there is a lack of clarity on wheither post-hoc corrections were applied. Please describe if this was the case, and if yes, which ones were used.

Results: seems fair and well-written. 

Discussion: In general, the discussion is well-written and professional. Nonetheless, some concerns are need to be addressed. Please synchronize with the introduction since it gives a feeling to the reader as if the same things are being repeated.

I think the therapeutic implications are important to be emphasized, though without any overstatement. 

This study, as any other study, has certain limitations. It is a good practice to acknowledge one's own limitations. Therefore it would be very important to add an ultimate discussion paragraph where all the study limitations would be addressed.

Author Response

We greatly appreciate reviewer’s valuable comments and suggestions. In response, we have modified the text and addressed all points raised by the reviewers. Our point-by-point responses to the reviewer’s comments are provided below. Each original comment is presented, followed by our “RESPONSE”. A marked manuscript with changes highlighted in red has been provided for your reference.

Reviewer comments:

I congratulate the authors for the article. It is both relevant and of scientific significance.

Please find below my specific comments. 

Q1. Introduction: it is well written and elaborated until the last paragraph. The ultimate introduction paragraph seems to be accidentally copy-pasted from the conclusion part!!! Therefore this paragraph needs to be deleted. Instead, another paragraph should be added. This new (final introduction) paragraph should provide information regarding the study aims and hypothesis.

RESPONSE: We sincerely thank the reviewer’s careful reading and constructive feedback. We acknowledge the oversight in the final paragraph of the Introduction. This paragraph has now been removed. In its place, we have added a new paragraph that clearly outlines the study aims and hypotheses, ensuring a more coherent and focused introduction. Please refer to line 76-84 in the revised manuscript.

Q2. Methods: this section is well-written, comprehensive, and reproductious. Nonetheless, I think a power analysis is needed to be conducted in order to describe if the study sample. 

Additionally, ANOVA is mentioned as the statistical test used to compare groups. Yet, there is a lack of clarity on wheither post-hoc corrections were applied. Please describe if this was the case, and if yes, which ones were used.

RESPONSE: We sincerely appreciate the reviewer's suggestion to conduct a power analysis to further validate the adequacy of our sample size. We acknowledge that power analysis is an important tool for ensuring the robustness of statistical conclusions. The lack of a formal power analysis is one of limitations of this study and we have discussed this issue in a special paragraph in the revised manuscript. Please refer to line 266-271 in the revised manuscript. Additionally, we have clarified the use ofpost-hoc Tukey’s testin the ANOVA analysis for multiple comparisons, and this information has now been explicitly stated in the Methods section and figure legends. Please refer to line 128-129, 150-151,156-157, 174-175, 349 in the revised manuscript.

Q3. Results: seems fair and well-written. 

RESPONSE: We are grateful for the reviewer’s acknowledgment of the Results section.

Q4. Discussion: In general, the discussion is well-written and professional. Nonetheless, some concerns are need to be addressed. Please synchronize with the introduction since it gives a feeling to the reader as if the same things are being repeated.

I think the therapeutic implications are important to be emphasized, though without any overstatement. 

This study, as any other study, has certain limitations. It is a good practice to acknowledge one's own limitations. Therefore it would be very important to add an ultimate discussion paragraph where all the study limitations would be addressed.

RESPONSE: We thank the reviewer for their valuable input on the Discussion section. We have carefully revised the Discussion to ensure it aligns more closely with the Introduction, avoiding unnecessary repetition. Furthermore, we have emphasized the therapeutic implications of our findings while maintaining a balanced and cautious tone. Please refer to line 358-364 in the revised manuscript. Lastly, as suggested, we have added a new paragraph at the end of the Discussion to explicitly address the limitations of the study, providing a more comprehensive and transparent discussion of our work. Please refer to line 252-270 in the revised manuscript.

Reviewer 2 Report

Comments and Suggestions for Authors

ijms-3474814

Title: The loss of Hjv aggravates muscle atrophy by promoting the activation of the TβRII/Smad3 pathway
Authors: Lu Wang, Wuchen Tao, Jiajie Jia, Min Yuan, Wenjiong Li, Peng Zhang*, Xiaoping Chen*

This study addresses an important issue in muscle physiology, particularly the role of Hemojuvelin (HJV) in muscle atrophy, which has not been extensively explored. It provides a mechanistic link between HJV loss and TβRII/Smad3 pathway activation, enhancing the understanding of signaling pathways involved in muscle atrophy. However, several aspects need further clarification, and various additional corrections have been identified in some parts of the paper, making further revisions necessary.

[Major concerns]

  1. The study does not fully explore whether iron dysregulation, a key function of HJV, plays a role in the observed atrophy. It should be clarified whether experiments related to this have been conducted, or if these aspects should be discussed in the Discussion section.
  2. While TβRII/Smad3 pathway activation is demonstrated, upstream regulatory mechanisms (e.g., how HJV directly influences TβRII) are not thoroughly investigated. It should be clarified whether experiments related to this have been conducted, or if these aspects should be discussed in the Discussion section.
  3. English: The English paper is well-written, but a few aspects need to be checked and corrected:
  • Consistent use of key terminology. Examples: TβRII at Line 21 vs. TβR II at Line 28 and more; TGF β1 at Line 314 vs TGF-β1 at Line 318; etc.
  • Errors in the notation of genes and proteins. HJV gene at Line 168; Hjv mRNA at Line 169; Hjv protein at Line 177, etc.
  1. Abbreviations: The use of abbreviations when writing a paper has many advantages besides simplicity of expression. To use an abbreviation, first write the abbreviation in parentheses after the full name, and then use the abbreviation from Introduction to the final Conclusion. Abbreviations should only be used if they are repeatedly used and if they are not used again, only the full name should be used. In particular, because of the characteristics of IJMS, where Materials and Methods is arranged at the end of the paper, the original words and abbreviations are written in the order they are used from the introduction, and only when the abbreviation is used repeatedly, the abbreviation can be used until the conclusion.
  2. In cases where abbreviations are used within figures or tables, please list these abbreviations along with their corresponding full names in the figure legends or at the bottom of corresponding tables. If there are two or more abbreviations, arrange them in alphabetical order. In this case, non-proper nouns should not have their first letters capitalized.
  3. Materials and Methods section - When naming a particular chemical company, you must provide location information such as company name, city, and/or state (abbreviation in the USA and Canada) and country. Once you have named a company with the information, you should only mention a company’s name thereafter.
  4. References: This paper cites a total of 29 references, but surprisingly, most of them are quite outdated. Given that since the WHO assigned a disease code for sarcopenia in 2016, a vast number of studies have been published not only on biomarker discovery but also on the development of new therapeutic drugs, it is unacceptable to rely on only a small number of old references. Therefore, try to cite more recent published articles for the references.

[Minor concerns]

  1. Line 17: Define WT at the abstract.
  2. Line 43: Define MyLC and MyHC.
  3. Line 51: Define AKT/mTOR.
  4. Line 92: ‘Western blotting analysis’ should be written as ‘Western blot analysis’.
  5. Line 98: Define CSA.
  6. Line 129: Define shRNA.
  7. References: Adjust the citation style of the references to conform to the IJMS guidelines, and ensure that any missing page numbers are accurately included. Examples: 3, 4, 7, 9, 10, 14, 17, 19, 21, 28, etc.

Overall, the manuscript can be considered to publication after major revision as indicated above.

Comments on the Quality of English Language

ijms-3474814

Title: The loss of Hjv aggravates muscle atrophy by promoting the activation of the TβRII/Smad3 pathway
Authors: Lu Wang, Wuchen Tao, Jiajie Jia, Min Yuan, Wenjiong Li, Peng Zhang*, Xiaoping Chen*

This study addresses an important issue in muscle physiology, particularly the role of Hemojuvelin (HJV) in muscle atrophy, which has not been extensively explored. It provides a mechanistic link between HJV loss and TβRII/Smad3 pathway activation, enhancing the understanding of signaling pathways involved in muscle atrophy. However, several aspects need further clarification, and various additional corrections have been identified in some parts of the paper, making further revisions necessary.

[Major concerns]

  1. The study does not fully explore whether iron dysregulation, a key function of HJV, plays a role in the observed atrophy. It should be clarified whether experiments related to this have been conducted, or if these aspects should be discussed in the Discussion section.
  2. While TβRII/Smad3 pathway activation is demonstrated, upstream regulatory mechanisms (e.g., how HJV directly influences TβRII) are not thoroughly investigated. It should be clarified whether experiments related to this have been conducted, or if these aspects should be discussed in the Discussion section.
  3. English: The English paper is well-written, but a few aspects need to be checked and corrected:
  • Consistent use of key terminology. Examples: TβRII at Line 21 vs. TβR II at Line 28 and more; TGF β1 at Line 314 vs TGF-β1 at Line 318; etc.
  • Errors in the notation of genes and proteins. HJV gene at Line 168; Hjv mRNA at Line 169; Hjv protein at Line 177, etc.
  1. Abbreviations: The use of abbreviations when writing a paper has many advantages besides simplicity of expression. To use an abbreviation, first write the abbreviation in parentheses after the full name, and then use the abbreviation from Introduction to the final Conclusion. Abbreviations should only be used if they are repeatedly used and if they are not used again, only the full name should be used. In particular, because of the characteristics of IJMS, where Materials and Methods is arranged at the end of the paper, the original words and abbreviations are written in the order they are used from the introduction, and only when the abbreviation is used repeatedly, the abbreviation can be used until the conclusion.
  2. In cases where abbreviations are used within figures or tables, please list these abbreviations along with their corresponding full names in the figure legends or at the bottom of corresponding tables. If there are two or more abbreviations, arrange them in alphabetical order. In this case, non-proper nouns should not have their first letters capitalized.
  3. Materials and Methods section - When naming a particular chemical company, you must provide location information such as company name, city, and/or state (abbreviation in the USA and Canada) and country. Once you have named a company with the information, you should only mention a company’s name thereafter.
  4. References: This paper cites a total of 29 references, but surprisingly, most of them are quite outdated. Given that since the WHO assigned a disease code for sarcopenia in 2016, a vast number of studies have been published not only on biomarker discovery but also on the development of new therapeutic drugs, it is unacceptable to rely on only a small number of old references. Therefore, try to cite more recent published articles for the references.

[Minor concerns]

  1. Line 17: Define WT at the abstract.
  2. Line 43: Define MyLC and MyHC.
  3. Line 51: Define AKT/mTOR.
  4. Line 92: ‘Western blotting analysis’ should be written as ‘Western blot analysis’.
  5. Line 98: Define CSA.
  6. Line 129: Define shRNA.
  7. References: Adjust the citation style of the references to conform to the IJMS guidelines, and ensure that any missing page numbers are accurately included. Examples: 3, 4, 7, 9, 10, 14, 17, 19, 21, 28, etc.

Overall, the manuscript can be considered to publication after major revision as indicated above.

Author Response

We greatly appreciate reviewer’s valuable comments and suggestions. In response, we have modified the text and addressed all points raised by the reviewers. Our point-by-point responses to the reviewer’s comments are provided below. Each original comment is presented, followed by our “RESPONSE”. A marked manuscript with changes highlighted in red has been provided for your reference.

Reviewer comments:

Title: The loss of Hjv aggravates muscle atrophy by promoting the activation of the TβRII/Smad3 pathway
Authors: Lu Wang, Wuchen Tao, Jiajie Jia, Min Yuan, Wenjiong Li, Peng Zhang*, Xiaoping Chen*

This study addresses an important issue in muscle physiology, particularly the role of Hemojuvelin (HJV) in muscle atrophy, which has not been extensively explored. It provides a mechanistic link between HJV loss and TβRII/Smad3 pathway activation, enhancing the understanding of signaling pathways involved in muscle atrophy. However, several aspects need further clarification, and various additional corrections have been identified in some parts of the paper, making further revisions necessary.

[Major concerns]

Q1. The study does not fully explore whether iron dysregulation, a key function of HJV, plays a role in the observed atrophy. It should be clarified whether experiments related to this have been conducted, or if these aspects should be discussed in the Discussion section.

RESPONSE: We appreciate the reviewer’s insightful comment regarding the role of iron dysregulation in the observed atrophy. While our study did not directly investigate iron dysregulation, we agree that this is an important aspect related to HJV function. We have now included a discussion of this topic in the revised manuscript, highlighting it as a potential area for future research. Please refer to line 252-266 in the revised manuscript.

Q2. While TβRII/Smad3 pathway activation is demonstrated, upstream regulatory mechanisms (e.g., how HJV directly influences TβRII) are not thoroughly investigated. It should be clarified whether experiments related to this have been conducted, or if these aspects should be discussed in the Discussion section.

RESPONSE: We thank the reviewer for raising this important point. Our study focused on demonstrating the activation of the TβRII/Smad3 pathway but did not explore the upstream regulatory mechanisms in detail. We have clarified this in the revised manuscript and have added a discussion on potential mechanisms by which HJV might influence TβRII. Please refer to line 220-223 in the revised manuscript.

Q3. English: The English paper is well-written, but a few aspects need to be checked and corrected: Consistent use of key terminology. Examples: TβRII at Line 21 vs. TβR II at Line 28 and more; TGF β1 at Line 314 vs TGF-β1 at Line 318; etc.

Errors in the notation of genes and proteins. HJV gene at Line 168; Hjv mRNA at Line 169; Hjv protein at Line 177, etc.

RESPONSE: We apologize for the inconsistencies in terminology and notation. We have carefully reviewed the manuscript and standardized the use of key terms, including TβRII, TGF-β1, and HJV (gene and protein). All instances have been corrected to ensure consistency throughout the text. Please refer to line 23, 25, 26, 31, 182, 191, 358, 362 in the revised manuscript.

Q4. Abbreviations: The use of abbreviations when writing a paper has many advantages besides simplicity of expression. To use an abbreviation, first write the abbreviation in parentheses after the full name, and then use the abbreviation from Introduction to the final Conclusion. Abbreviations should only be used if they are repeatedly used and if they are not used again, only the full name should be used. In particular, because of the characteristics of IJMS, where Materials and Methods is arranged at the end of the paper, the original words and abbreviations are written in the order they are used from the introduction, and only when the abbreviation is used repeatedly, the abbreviation can be used until the conclusion.

In cases where abbreviations are used within figures or tables, please list these abbreviations along with their corresponding full names in the figure legends or at the bottom of corresponding tables. If there are two or more abbreviations, arrange them in alphabetical order. In this case, non-proper nouns should not have their first letters capitalized.

RESPONSE: We thank the reviewer for the detailed guidance on the use of abbreviations. We have revised the manuscript to ensure that all abbreviations are introduced correctly at their first mention and used consistently thereafter. Additionally, we have included the abbreviations used in figures and tables in the corresponding legends, arranged alphabetically and formatted according to the reviewer’s suggestions. Please refer to line 19, 47, 48, 55, 62, 66, 80, 81, 107, 129, 139, 151, 157, 175 in the revised manuscript.

Q5. Materials and Methods section - When naming a particular chemical company, you must provide location information such as company name, city, and/or state (abbreviation in the USA and Canada) and country. Once you have named a company with the information, you should only mention a company’s name thereafter.

RESPONSE: We appreciate the reviewer’s attention to detail. We have updated the Materials and Methods section to include the full location information (company name, city, state, and country) for all chemical reagents and supplies. Subsequent mentions of these companies now use only the company name, as suggested. Please refer to line 276, 279, 295-301, 324-335, 343 in the revised manuscript.

Q6. References: This paper cites a total of 29 references, but surprisingly, most of them are quite outdated. Given that since the WHO assigned a disease code for sarcopenia in 2016, a vast number of studies have been published not only on biomarker discovery but also on the development of new therapeutic drugs, it is unacceptable to rely on only a small number of old references. Therefore, try to cite more recent published articles for the references.

RESPONSE: We acknowledge the reviewer’s concern regarding the outdated references. We have carefully reviewed the literature and updated the reference list to include more recent studies. This has strengthened the context and relevance of our work. Please refer to line 464-486 in the revised manuscript.

Q7. [Minor concerns]

Line 17: Define WT at the abstract.

Line 43: Define MyLC and MyHC.

Line 51: Define AKT/mTOR.

Line 92: ‘Western blotting analysis’ should be written as ‘Western blot analysis’.

Line 98: Define CSA.

Line 129: Define shRNA.

RESPONSE: We have addressed all the minor concerns raised by the reviewer. Defined “WT” in the abstract. Please refer to line 19. Defined “MyLC,” “MyHC,” “AKT/mTOR,” “CSA,” and “shRNA” at their first mention. Please refer to line 47, 55, 107, 139. Corrected “Western blotting analysis” to “Western blot analysis.”Please refer to line 99 in the revised manuscript.

Q8. References: Adjust the citation style of the references to conform to the IJMS guidelines, and ensure that any missing page numbers are accurately included. Examples: 3, 4, 7, 9, 10, 14, 17, 19, 21, 28, etc.

RESPONSE: We have adjusted the citation style of references to conform to IJMS guidelines, including adding missing page numbers where applicable. Please refer to line 390, 394, 396, 401, 403, 406, 413, 420, 434, 439, 444, 457 in the revised manuscript.

Overall, the manuscript can be considered to publication after major revision as indicated above.

We sincerely thank reviewer’s constructive feedback, which has significantly improved the quality of our manuscript. We hope that the revisions address all concerns and meet the journal’s standards for publication.

Round 2

Reviewer 2 Report

Comments and Suggestions for Authors

I recommend the acceptance of the paper as the issues, including those pointed out during the first review process, have been properly corrected.